# UNCERTAINTY DISTILLATION: TEACHING LANGUAGE MODELS TO EXPRESS SEMANTIC CONFIDENCE

## ABSTRACT

As large language models (LLMs) are increasingly used for factual question-answering, it becomes more important for LLMs to have the capability to communicate the likelihood that their answer is correct. For these verbalized expressions of uncertainty to be meaningful, they should reflect the error rates at the expressed level of confidence. However, when prompted to express confidence, the error rates of current LLMs are inconsistent with their communicated confidences, highlighting the need for uncertainty quantification methods. Many prior methods calculate *lexical* uncertainty, estimating a model's confidence in the specific string it generated. In some cases, however, it may be more useful to estimate *semantic* uncertainty, or the model's confidence in the answer regardless of how it is verbalized. We propose a simple procedure, **uncertainty distillation**, to teach an LLM to verbalize calibrated semantic confidences. Using held-out data to map initial uncertainty estimates to meaningful probabilities, we create examples annotated with verbalized probabilities for supervised fine-tuning. We find that our method yields verbalized confidences that correlate well with observed error rates, even when compared to strong baselines, some of which are more than twenty times slower at inference time.

## 1 INTRODUCTION

Advances in LLM research have led to instruction-tuned generative models with impressive capabilities on many challenging tasks (OpenAI et al., 2024; Jiang et al., 2023; Dubey et al., 2024). While the flexibility and quality of these models is appealing, they may still hallucinate or give incorrect answers (Rawte et al., 2023; Bai et al., 2024). However, language models do not readily provide an interpretable measure of a model's likelihood of correctness. LLMs tend to produce poorly-calibrated confidences when prompted to do so, and are often confidently incorrect (Xiong et al., 2024). Furthermore, the elicited confidences may be impacted in unexpected ways by the choice of prompt (Sclar et al., 2023), such as the interpretation of "very confident" being dependent on the wording of the prompt.

There are several other approaches as an alternative to prompting. Models' token-level probabilities can be used to provide information as a measure of *lexical* uncertainty, which gives information about the likelihood of a generated string. This is often useful; however, the same fact can be expressed in any number of ways—"Berlin's the capital of Germany" or "The capital of Germany is Berlin!" or "Die Hauptstadt Deutschlands ist Berlin"—all capturing the same meaning (Kuhn et al., 2023). *Semantic* uncertainty is therefore challenging to capture, as token-level probabilities are influenced by the phrasing of an answer just as much as the semantics of the answer itself. This issue is particularly challenging for models employing large vocabularies such as multilingual language models, language models employing byte or character-level tokenization, or when using LLMs that are prone to producing extraneous outputs (Xue et al., 2021; Wang et al., 2024a).

We present *uncertainty distillation*[1], a scheme for fine-tuning a language model to verbalize uncertainty based on its own internal state. Notably, uncertainty distillation teaches models to estimate their semantic—rather than lexical—uncertainty, as the distilled confidences are estimated from the

---

[1]We choose this name to evoke *model* distillation, a process which like uncertainty distillation requires an offline cost to generate data to train a more efficient model.

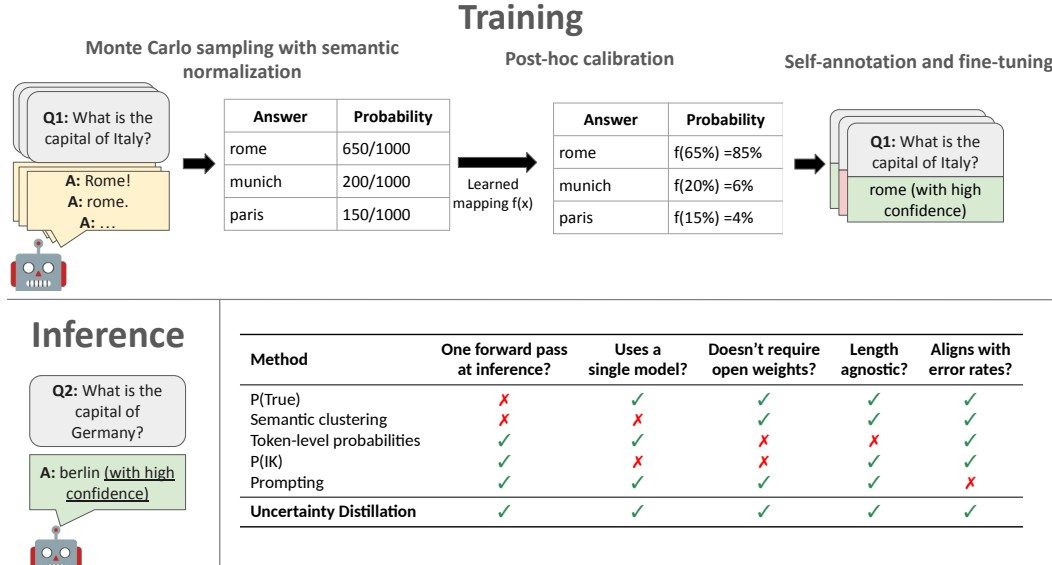

Figure 1: An overview of our method, Uncertainty Distillation. At training time, in **Monte Carlo sampling with semantic normalization**, we sample repeatedly from our language model, and use a normalization function to consolidate answers with the same semantic meaning. By consolidating the counts, we obtain a Monte Carlo estimate of each answer's probability. In **post-hoc calibration**, we pass this estimate through a learned post-hoc calibration function to better align it with its likelihood of correctness. Finally, in **self-annotation and fine-tuning**, we translate these probabilities to verbalized signifiers and fine-tune a model to output verbalized confidences in addition to the answer. This method confers several advantages, listed in the table: at inference time, a single model generates the confidence efficiently in a single pass, providing high discriminative power with little computational overhead. The length of the answer does not directly impact the confidence, and white-box access to weights is not required.

probabilities of semantically normalized outputs, rather than relying on token-level probabilities. At inference time, models trained using uncertainty distillation efficiently generate a well-calibrated and interpretable statement of confidence in their answers, such as "Berlin is the capital of Germany [high confidence]."[2] Our approach enables semantically equivalent but lexically different predictions to be assigned the same confidence, and a single generation with multiple claims can each be assigned different confidences. Uncertainty distillation is computationally inexpensive at inference time, generating only a handful of additional tokens. Compared to methods such as P(IK) (Farquhar et al., 2024), we do not require a separate uncertainty network; our approach uses standard supervised fine-tuning recipes for LLMs. Our method can be applied to open-source LLMs as well as proprietary LLMs that allow fine-tuning; white-box access to model weights is not required.

*Uncertainty distillation* involves self-annotation of any desired QA dataset with the base model's calibrated uncertainties, which are then used to fine-tune that model to produce verbalized confidences. At a high level (Figure 1), our approach consists of three steps: (1) obtaining semantic uncertainty estimates from the model; (2) post-hoc calibrating these into meaningful probabilities; and (3) teaching the model via supervised fine-tuning to output verbalized confidences along with its predictions.

**Summary of contributions**

- We propose uncertainty distillation, a simple yet effective scheme which uses supervised fine-tuning to teach LLMs to output calibrated semantic confidence statements along with their predictions. We publish our code and trained models.[3]

---

[2]The uncertainty could be expressed in a variety of ways, including using special characters or numeric values.

[3]https://anonymous.4open.science/r/uncertainty-distillation-anon-05CB/README.md

- We demonstrate that uncertainty distillation achieves easily interpretable results and compares favorably to several powerful baselines.
- We analyze whether models trained with uncertainty distillation can apply their representations of uncertainty to unseen topics at inference time without further fine-tuning.

## 2 RELATED WORK

**Linguistic calibration and verbalized confidences**   Generally, calibration refers to the concept that predicted probabilities should align with the probability of correctness (Guo et al., 2017). Mielke et al. (2022) additionally propose the conception of "linguistic calibration"—that models demonstrate uncertainty or doubt through natural language when they are incorrect, determining this uncertainty by using a predictor to determine the likelihood that an answer is correct and considering that to be the model's uncertainty. There are significant advantages to verbalizing uncertainty: for one, there is relatively low computational overhead to generate several extra tokens, while using a separate calibration model to estimate confidence and then communicate this information to the user requires more computation at inference time (Yang et al., 2024). Verbalized confidences are also readily interpretable to an LLM when reasoning about uncertainty, or to an average end-user regardless of experience or background.

**Lexical uncertainty quantification**   Lexical uncertainty quantification metrics using information from token-level probabilities are commonly used and frequently effective (Hu et al., 2023; Malinin & Gales, 2021). These probabilities are easily obtainable, do not require additional inference-time compute to generate, and often provide sufficient information for downstream use cases: e.g. error correction in chain of thought (Yin et al., 2024), hallucination detection (Arteaga et al., 2024), or out-of-distribution data detection (Hendrycks et al., 2020b). However, there are several disadvantages to lexical uncertainty quantification: it relies on model probabilities which may not be well-calibrated (Guo et al., 2017), and is often ineffective on calculating uncertainty of long generations (Zhang et al., 2024). The latter, in particular, may present problems for end users, as models trained using Reinforcement Learning from Human Feedback (RLHF) are often incentivized to produce long outputs (Singhal et al., 2024). It is therefore important to consider uncertainty quantification methods that do not rely on token-level probabilities to estimate uncertainty.

**Semantic uncertainty quantification**   In contexts where lexical uncertainty falls short, a natural method to obtain verbalized confidences might be to simply prompt a model to output confidences, providing an estimate of uncertainty without explicitly using token-level probabilities. However, in practice, LLMs tend to overestimate their own confidence, possibly because human annotators tend to prefer texts with fewer markers of uncertainty (Zhou et al., 2024). This, in turn, suggests while simply altering prompts may result in improved confidence estimates (Xiong et al., 2024; Tian et al., 2023), models may be fundamentally limited in their ability to acknowledge uncertainty without further training.

Running multiple steps at inference time may provide a better estimate of semantic probability. Xiong et al. (2024) investigate several inference-time strategies which use multiple steps to estimate model uncertainty, such as sampling several answers on the same question or noting if a model changes its answer when prompted with a misleading alternative. While these methods do lead to improvements in LLM calibration, no single intervention consistently emerges as the most successful, and the authors note there is significant scope for improvement. Kuhn et al. (2023) and Farquhar et al. (2024) more explicitly relate this to semantic uncertainty, and find that sampling $m$ predictions from the model and clustering by semantic equivalence results in a robust measure of semantic uncertainty that compares favorably to lexical uncertainty. A major disadvantage of these sampling-based approaches is their increased computational complexity at inference time, however; for instance, the semantic clustering approach of Farquhar et al. (2024), which we compare to in our experiments, requires 20 samples and calls to a separate entailment model at inference time.

## 3 METHOD

We propose a simple training recipe, illustrated in Figure 1 and described below, to allow a language model to express confidences that correlate with expected error rates on held-out data.

### 3.1 MONTE CARLO SAMPLING WITH SEMANTIC NORMALIZATION

Assuming input $x$ and output $y$, we are looking to find $\sum_{y \in Y_{\text{equivalent}}} P(y \mid x)$, the model's likelihood of producing this answer or one that is semantically equivalent; however this would require marginalization over an infinite set of strings $Y$. To make this a tractable problem, we use a Monte Carlo approximation, where our estimate of the models' predictive distribution improves with $N$, at the expense of additional offline computation. Note however that we do not assume this quantity is a meaningful probability out-of-the-box due to potential overfitting or underfitting of the base model. To diagnose potential miscalibration of the base model as well as correct for it, we may fit a post-hoc calibrator if the training data demonstrates miscalibration.

In more detail, to fit a post-hoc calibrator, we need a supervised dataset of datapoints not seen at training time $\{X^{\text{cal}}, Y^{\text{cal}}\}$. For each example $x \in X^{\text{cal}}$ we sample $N$ candidate answers $\{\hat{y}_i\}_{i=1}^{N} \sim P_{\theta}(Y \mid X = x)$ from a model's predictive distribution[4]. Before calculating the relative frequency of strings, we apply a normalization function (or set of normalization functions) to consolidate semantically similar outputs. In the short-form QA tasks we consider in §4, we use the simple normalization function of isolating a multiple choice answer using tags, removing punctuation and standardizing capitalization; we demonstrate how semantic normalization can be applied to more complex tasks in Appendix A. After consolidating strings belonging to the same event, the relative frequency $f$ of these events is a measure of the LLM's uncertainty in those events, although this may not be a well-calibrated probability.

### 3.2 POST-HOC CALIBRATION

Neural networks are prone to miscalibration. A common remedy is to apply *post-hoc* calibration methods, which usually involve some form of regression on predicted scores to transform them into meaningful probabilities. Specifically, we post-hoc calibrate the relative frequencies of each semantic cluster found in the previous step. Two common options for post-hoc calibration are isotonic regression and Platt scaling (sometimes called temperature scaling) (Guo et al., 2017). Our approach uses a model's predictions on $\{X^{\text{cal}}, Y^{\text{cal}}\}$ to diagnose and mitigate badly-calibrated initial model probabilities. We fit an isotonic regression model[5] on our calibration set by comparing the predicted scores to observed labels.[6] We compare each prediction $\hat{y}$ with score $f$ to observed events $y$. This yields a calibration map $c : \mathbb{R} \to [0, 1]$ we apply to the relative frequencies of events from samples in the previous step to yield probabilities.

### 3.3 SELF-ANNOTATION AND FINE-TUNING

We compute the calibrated probability $p = c(f)$ associated with each prediction in the held-out calibration data, and choose a mapping into discrete confidence bins. Several options are possible for this binning function $b$, including adaptive schemes as well as uniform schemes, the number of bins $B$, and so on. In our experiments, we focus on a simple fixed-width scheme with 5 bins. Let $\hat{Y}$ denote the set of all predictions on $X^{\text{cal}}$, and, if the model was previously fine-tuned on a supervised training set $X^{\text{train}}$, we include predictions on $X^{\text{train}}$. We transform each prediction and calibrated confidence into a training example for a round of supervised fine-tuning by verbalizing the corresponding bin in the answer. For example, the fifth of five bins may correspond to "very high confidence." The token sequences chosen to encode each bin are arbitrary, as we discuss in Appendix F; for easy interpretability, we use short confidence descriptors in this paper, namely "very low," "low," "medium," "high," and "very high."

In our scheme, we simply append the verbalized confidence to all answers. For instance, if the model generates 900 correct answers and 100 incorrect answers, there are two available data points that could potentially be added to the dataset:

---

[4]This model may have been fine-tuned on the specific task as in Appendix C or instruction-tuned as in §4 and Appendix B.

[5]We use isotonic regression for ease of training and use; this could be replaced with a different post-hoc calibration method, or omitted entirely as discussed in Appendix G. We use the `scikit-learn 1.5.2` with no modification.

[6]We discuss the effect of post-hoc calibration further in Appendix G.

```
<correct answer> (with very high confidence)

<incorrect answer> (with very low confidence)
```

While correct answers should be added as training data, appending the confidence scores to *incorrect* answers may improve the model's ability to correctly verbalize its own confidence. However, it may also decrease the accuracy of the QA model. We introduce a hyperparameter to control the number of incorrect answers added to the training data. In §B.2, we further investigate the impact of this hyperparameter.

Starting from the sampled model, we perform supervised fine-tuning on these self-annotated targets with verbalized confidences to estimate a second model capable of verbalizing its confidence. If training an instruction-tuned model, we append an additional instruction such as "Additionally state how confident you are in your answer." to the preexisting instruction[7]. If a reasoning trace has been generated during sampling, we randomly select a reasoning trace to add to the target answer from all possible options. At inference time, we obtain predictions *and verbalized confidences* from this new model on held-out test data. This test data has no overlap with the post-hoc calibrated training set, and can even be drawn from an entirely different dataset, as in §6. We remark that our model incurs little additional cost at inference time, as opposed to other confidence elicitation methods which require inference-time sampling (Farquhar et al., 2024; Xiong et al., 2024).

## 4  EXPERIMENTAL SETUP

| DATASET | MODEL | METHOD | AUROC | ACC | HIGH ACC | HIGH % |
|---|---|---|---|---|---|---|
| MMLU | MINISTRAL-8B | UD (OURS) | **0.693** | 0.601 | 0.766 | 49.7 |
| | | LEXICAL BASELINE | 0.627 | 0.551 | 0.555 | 99.2 |
| | | PROMPTING | 0.587 | **0.637** | 0.643 | 97.4 |
| | | P(IK) | 0.670 | 0.566 | 0.639 | 83.1 |
| | | P(TRUE) | 0.471 | 0.585 | 0.583 | 96.6 |
| | | SEM. CLUSTERING | 0.667 | 0.577 | **0.821** | 34.6 |
| | LLAMA-3B | UD (OURS) | **0.743** | 0.532 | **0.759** | 42.4 |
| | | LEXICAL BASELINE | 0.644 | 0.511 | 0.600 | 62.0 |
| | | PROMPTING | 0.548 | **0.613** | 0.647 | 73.9 |
| | | P(IK) | 0.692 | 0.567 | 0.688 | 59.8 |
| | | P(TRUE) | 0.550 | 0.554 | 0.558 | 98.6 |
| | | SEM. CLUSTERING | 0.646 | 0.560 | 0.727 | 63.8 |
| SOCIALIQA | MINISTRAL-8B | UD (OURS) | 0.671 | 0.713 | **0.792** | 53.7 |
| | | LEXICAL BASELINE | 0.600 | **0.738** | 0.760 | 85.7 |
| | | PROMPTING | 0.539 | 0.721 | 0.738 | 95.8 |
| | | P(IK) | **0.676** | 0.650 | 0.713 | 85.0 |
| | | P(TRUE) | 0.491 | 0.712 | 0.710 | 92.5 |
| | | SEM. CLUSTERING | 0.603 | 0.659 | 0.780 | 17.7 |
| | LLAMA-3B | UD (OURS) | **0.784** | 0.653 | 0.833 | 55.1 |
| | | LEXICAL BASELINE | 0.531 | 0.673 | 0.687 | 95.3 |
| | | PROMPTING | 0.545 | **0.685** | 0.712 | 67.2 |
| | | P(IK) | 0.669 | 0.664 | **0.839** | 26.4 |
| | | P(TRUE) | 0.505 | 0.681 | 0.682 | 99.1 |
| | | SEM. CLUSTERING | 0.601 | 0.675 | 0.758 | 34.0 |

Table 1: Binned AUROC and accuracy metrics for our large models and datasets. We find that uncertainty distillation (UD) leads to increased AUROC and accuracy in high-confidence categories. `Accuracy` is the overall accuracy, and `High Accuracy` is the accuracy for the most confident predictions. We find that uncertainty distillation with one generation achieves similar or improved `High Accuracy` compared to other methods, including those using multiple samples.

We examine the efficacy of uncertainty distillation in two settings. First, we demonstrate the success of uncertainty quantification with large language models trained on several standard QA bench-

---
[7]See Appendix E for details on the specific prompts used in each experiment.

marks. Second, we examine whether the models can still accurately forecast uncertainty when applied to datasets not seen during uncertainty distillation.

## 4.1 UNCERTAINTY DISTILLATION IN-DOMAIN

**Datasets** We demonstrate uncertainty distillation using two multiple-choice question answering datasets, the Massive Multitask Language Understanding benchmark (MMLU) (Hendrycks et al., 2020a) and the Social Interaction Question Answering dataset (SocialIQA) (Sap et al., 2019). MMLU consists of multiple choice questions over 57 subjects such as high school psychology or formal logic. We take a subset of 20,000 questions from the training set to act as our calibration data, a subset of 500 questions from the validation set to act as our validation data, and a subset of 2,000 quesions from the test set to act as our test data. SocialIQA is a dataset consisting of question/answer pairs about social situations. We take a subset of 20,000 questions from the training set to act as our calibration data, a subset of 500 questions from the training set to act as our validation data, and use the existing validation split as our test data. For both datasets we set $N = 100$, i.e. we take 100 samples per question to construct our initial Monte Carlo estimate of confidence[8]

**Models and baselines** We validate uncertainty distillation on these datasets using two modern instruction-tuned LLMs, Llama-3.2-3B-Instruct (Dubey et al., 2024) and Ministral-8B-Instruct-2410 (Jiang et al., 2023). When performing uncertainty distillation with Ministral-8B, we use LoRA (Hu et al., 2021). For the `Lexical` baseline, we extract token-level probabilities from the language model on our training/calibration split[9] and use this to train an isotonic regression model to calibrate the average token-level probability for each answer.[10] To measure the model's ability to verbalize its confidence prior to uncertainty distillation , in `Prompting` we prompt the base model to output its own confidence in its answer. We report this baseline for these models, and discuss the prompts used in Appendix E. We also compare to `P(IK)` from Farquhar et al. (2024) which learns a mapping from hidden states to uncertainty scores, and `P(True)` from Kadavath et al. (2022). Finally, we compare to the `Semantic Clustering (SC)` approach from Farquhar et al. (2024). Both `P(True)` and `Semantic Clustering` generate 20 samples from the model to compute uncertainty scores, unlike our approach which uses a single generation.

## 4.2 UNCERTAINTY DISTILLATION UNDER DOMAIN SHIFTS

We have discussed uncertainty distillation as a method that allows a model to forecast its own certainty. However, one potential reason for its success is if it is instead learning information about the *dataset*, and is learning to associate low confidence with types of questions that it has previously gotten wrong.[11] By changing the evaluation dataset, we demonstrate that the representation of uncertainty is not limited to only the domain of the training dataset.

**Datasets** We use SocialIQA and MMLU as described above. We also evaluate our models on the 500 examples in the test split of OpenbookQA(Mihaylov et al., 2018), an elementary-level science multiple choice question answering dataset.

**Models and baselines** In this experiments, we use the models described in §4.1 without further fine-tuning. Models trained on MMLU are tested on SocialIQA and OpenbookQA; Models trained on SocialIQA are tested on MMLU and OpenbookQA. We compare to the `Lexical` and `P(IK)` baselines described above, as these are the only two methods that require supervised data (`Lexical` to fit a calibration map and `P(IK)` to train a regressor) and would be affected by domain shifts.

---

[8]We chose $N$ based on the small-scale experiments described in Appendix D.

[9]As we do not have an initial fine-tuning step, these are equivalent.

[10]We use the average probability rather than the sequence probability to normalize over different lengths, as Kuhn et al. (2023) find this improves performance.

[11]For instance, if models perform particularly poorly on chemistry questions, it might output low uncertainty only because the question uses words such as "hydrogen", rather than learning an innate representation of uncertainty.

### 4.3 METRICS

We report the area under the receiver operating characteristic curve (AUROC),[12] which represents the probability that a randomly chosen correct answer will be in a higher-confidence bin than a randomly chosen incorrect answer. This metric is well established in previous literature (see e.g., Hu et al. (2023)), and compares the relative rather than absolute probabilities, which allows us to use it effectively with discrete verbalized confidences.[13] Baseline methods that return a continuous score are binned to five categories to represent converting to a comparable verbalized confidence[14]. For all methods, we plot the percentage of accurate answers in each bin to examine if confidence corresponds well with accuracy. We also report overall model accuracy, to evaluate the tradeoff between accuracy and calibration. Finally, we report `high accuracy` (accuracy of predictions in "very high" and "high" bins) and `high %` (percentage of predictions in "very high" and "high" bins). As an established use-case for verbalized confidences is to reject lower-confidence predictions, this provides information about how useful the LLM's predictions in rejecting incorrect answers and preserving a high number of correct answers.[15]

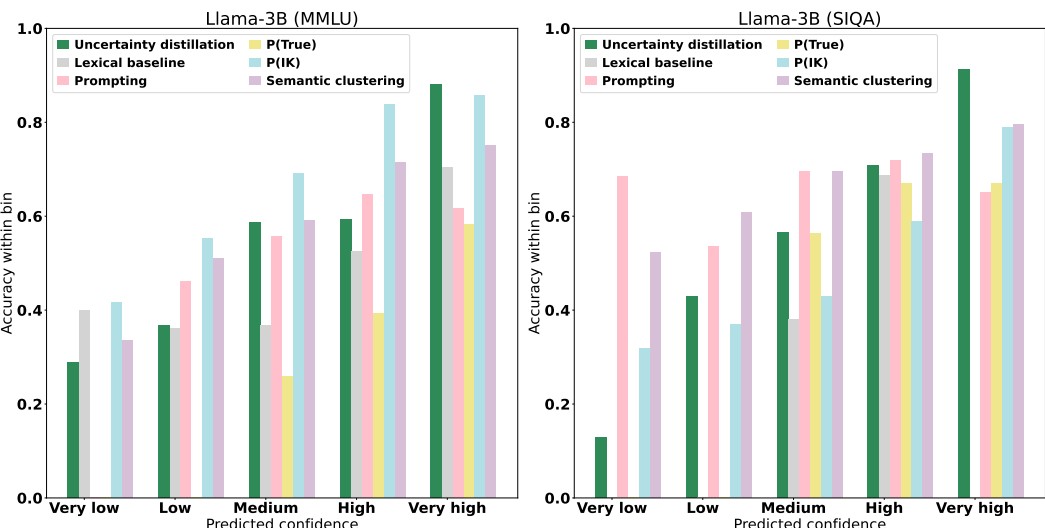

Figure 2: Average accuracy within each confidence bin for our experiments with Llama (Mistral results in Figure 6). We find that our confidence bins correspond well with accuracy within the bin, while our baselines may not exhibit similar correspondence. We do not plot bins with fewer than 10 samples.

## 5 RESULTS AND DISCUSSION

Figure 2 shows some of our results comparing uncertainty distillation to the lexical uncertainty baseline in terms of average accuracies in each confidence bin[16]. In plots like this, an ideal model would exhibit a diagonal trend line where outputs reported to have high confidence indeed have high accuracy, and those in the low confidence bins have lower accuracy. We find that the verbalized confidences produced by uncertainty distillation are highly *interpretable*, with high correspondence between accuracy of answers within a bin and that bins confidence. In contrast, confidence scores generated by the baselines may not correspond well with the actual accuracies within that bin. For

---

[12]Calculated using `scikit-learn` 1.5.2

[13]We do not report Expected Calibration Error (ECE), as it requires comparing a numerical probability to the prediction's true label, while our method and the semantic clustering baseline do not output numerical probabilities. Furthermore, many forms of calibration error require the choice of several hyperparameters such as binning strategy or regularization, which can have a large impact on performance (Nixon et al., 2019).

[14]If the probability is not normalized, we learn a binner using the range from validation data.

[15]The fact that high accuracy is not perfect also highlights a risk of confidence estimation: namely, that it increases trust in an answer that still may be incorrect.

[16]We present the remaining two settings in Appendix I.

instance, accuracy within the lowest confidence bin for the prompting baseline is 0.684 with Llama-3B on SocialIQA, while accuracy within the highest confidence bin is 0.651.

Table 1 summarizes these plots in terms of AUROC score. AUROC is consistently high with uncertainty distillation, generally outperforming other methods. We conclude that uncertainty distillation is effective for estimating confidence in an answer. AUROC is highest for uncertainty distillation for all experiments except Ministral-8B on SocialIQA, where it outperforms all baselines bu `P(IK)`. In particular, we note that uncertainty distillation consistently achieves higher AUROC than semantic clustering(Kuhn et al., 2023), despite semantic clustering requiring 20 samples and a computationally intensive clustering step at inference time: for instance, uncertainty distillation achieves AUROC of 0.784 with Llama-3B on SocialIQA, while semantic clustering achieves AUROC of 0.601.

The table also reports the accuracy of the highest confidence bin and the overall accuracy across all bins. While AUROC is the main metric for assessing performance, accuracy is also useful for understanding the nuances of the result. We find that uncertainty distillation does not lead to notable drops in overall accuracy, and that accuracy in the highest bins increases dramatically without restricting to drastically low amount high-confidence predictions (`High %` stays consistently above 40%). Uncertainty distillation achieves the best `High Accuracy` most cases. The exceptions are Ministral-8B on MMLU and Llama-3B on SocialIQA. In both these cases, the comparatively strong high accuracy results from notably smaller percentage of samples in high-confidence bins for these baselines, with only 34.6% of predictions being high-confidence for the baseline for MMLU compared to 49.7% for uncertainty distillation, and only 26.4% of predictions being high-confidence for the baseline for SocialIQA, compared to 55.1% for uncertainty distillation.

## 6 SUCCESS UNDER DOMAIN SHIFTS

Table 2 shows uncertainty distillation results compared to supervised baselines. We find that uncertainty distillation (UD) consistently achieves high AUROC despite the domain shifts, outperforming in all cases but Ministral-8B trained on SocialIQA and tested on OpenbookQA, which is outperformed by the lexical baseline and marginally by `P(IK)`.

In Table 2, we compare only to similarly out-of-domain baselines (i.e., also fit on data from a different distribution). A priori, one might expect that our approach fine-tuned for a specific dataset would significantly degrade in performance on a different dataset due to biases or spurious correlation. However, we find that out-of-domain uncertainty distillation outperforms all unsupervised baselines (semantic clustering, prompting, and `P(True)`), with the sole exception of Ministral-8B semantic clustering on MMLU. Notably, semantic clustering requires 20 samples from the language model, making uncertainty distillation more efficient at inference time by an order of magnitude. This result demonstrates that the representations of uncertainty learned by the model during uncertainty distillation are not limited to the training dataset, but can be applied to new datasets while still outperforming baselines unaffected by domain shifts.

## 7 UNCERTAINTY DISTILLATION WITH BLACK-BOX MODELS

Increasingly, large foundation models are not being released publicly, and even if they were, few groups posses the hardware to run large mixture-of-experts models efficiently. One advantage of uncertainty distillation is that it does not require open access to model weights; therefore, if there is an option to tune a model through an API, uncertainty distillation can still be used. Here, we demonstrate the success of uncertainty distillation in this case.

**Model and dataset** To strike a balance between cost and quality, we use Google's `gemini-2.5-flash-lite` model. Since this is a significantly more capable model than the open-weight models used elsewhere in this paper, we use a more challenging benchmark: MMLU-Pro (Wang et al., 2024b), a variant of MMLU designed to be more challenging and which includes a broader set of questions, including questions requiring reasoning. Note that the original MMLU dataset already covers a wide range of topics, so this benchmark helps us understand whether a single model can successfully estimate uncertainty across a wide range of settings, given only a a few

| Train Dataset | Test Dataset | Model | Method | AUROC | Acc |
|---|---|---|---|---|---|
| MMLU | SocialIQA | Ministral-8B | UD (Ours) | **0.657** | 0.676 |
| | | | Lexical Baseline | 0.593 | **0.738** |
| | | | P(IK) | 0.618 | 0.636 |
| | | Llama-3B | UD (Ours) | **0.717** | 0.627 |
| | | | Lexical Baseline | 0.574 | **0.670** |
| | | | P(IK) | 0.675 | 0.655 |
| | OpenbookQA | Ministral-8B | UD (Ours) | **0.757** | 0.734 |
| | | | Lexical Baseline | 0.676 | **0.812** |
| | | | P(IK) | 0.683 | 0.736 |
| | | Llama-3B | UD (Ours) | **0.834** | **0.733** |
| | | | Lexical Baseline | 0.647 | 0.680 |
| | | | P(IK) | 0.770 | 0.722 |
| SocialIQA | MMLU | Ministral-8B | UD (Ours) | **0.644** | **0.599** |
| | | | Lexical Baseline | 0.635 | 0.551 |
| | | | P(IK) | 0.605 | 0.553 |
| | | Llama-3B | UD (Ours) | **0.714** | 0.547 |
| | | | Lexical Baseline | 0.569 | 0.528 |
| | | | P(IK) | 0.687 | **0.572** |
| | OpenbookQA | Ministral-8B | UD (Ours) | 0.700 | 0.746 |
| | | | Lexical Baseline | **0.719** | **0.812** |
| | | | P(IK) | 0.704 | 0.718 |
| | | Llama-3B | UD (Ours) | **0.758** | **0.755** |
| | | | Lexical Baseline | 0.549 | 0.680 |
| | | | P(IK) | 0.693 | 0.694 |

Table 2: AUROC and accuracy metrics for Uncertainty Distillation (UD) tested on out-of-domain datasets compared to out-of-domain supervised baselines tested. Uncertainty distillation consistently achieve high AUROC on the novel test set in comparison to the supervised baselines, which are more inconsistent when dealing with domain shifts.

hundred demonstrations from each domain. We use an existing split of the data into training and evaluation sets, and we further split the evaluation set into 50% validation data and 50% test data[17].

**Baselines** The API restrictions preclude uncertainty estimation approaches that inspect model activations such as `P(IK)` or approaches that require next-token logits such as the lexical baseline. Nonetheless, we can fairly compare to baselines involving prompting or repeated sampling, so we include comparisons to prompting for verbalized confidences and semantic clustering. For semantic clustering, we include results for 8, 16, and 32 samples at inference time.

**Procedure** The procedure is identical using a commercial API or fine-tuning models locally. First, we generate 128 samples on the training split and then apply semantic clustering to estimate the relative frequency of each prediction. We then post-hoc calibrate the relative frequencies using either temperature scaling or isotonic regression. Finally, we create a fine-tuning dataset consisting of predictions and their calibrated confidences. On the validation data, we compare the performance of models trained with varying numbers of incorrect predictions, as described in §3.3 and Appendix L. The base model is then fine-tuned using LoRA via the Google Generative AI SDK[18], and this fine-tuned model is then used to make predictions on validation or test data.

**Results and analysis** The cost of running the entire pipeline was approximately $20, including generating samples, fine-tuning, and generating predictions on held-out data. For experimenting with different fine-tuning hyper-parameters, we could re-use samples, further controlling costs. We

---

[17]https://huggingface.co/datasets/answerdotai/MMLU-SemiPro
[18]https://docs.cloud.google.com/vertex-ai/generative-ai/docs/models/gemini-use-supervised-tuning

show results in Table 3, demonstrating that uncertainty distillation outperforms the other black-box methods. In particular, we note that at inference uncertainty distillation only requires a single pass, while semantic clustering requires eight to thirty-two.

| METHOD | AUROC | ACC | HIGH ACC | COST PER ANSWER |
|---|---|---|---|---|
| UD (OURS) | **0.762** | 0.490 | **0.706** | 1x |
| PROMPTING | 0.582 | 0.498 | 0.503 | 1x |
| SEM. CLUSTERING (8) | 0.713 | **0.508** | 0.562 | 8x |
| SEM. CLUSTERING (16) | 0.715 | 0.505 | 0.575 | 16x |
| SEM. CLUSTERING (32) | 0.718 | 0.505 | 0.581 | 32x |

Table 3: AUROC and accuracy metrics for the API-tuning experiments for `gemini-2.5-flash-lite`. We find that uncertainty distillation (UD) significantly outperforms all baselines in AUROC and high accuracy, and achieves similar accuracy to the only single-generation baseline, prompting. With the multi-generation baseline of semantic clustering increasing the number of samples to 32 does not cause semantic clustering to approach the performance of uncertainty distillation. We also note that semantic clustering costs 8-32x more than uncertainty distillation.

The limitations in applicable baselines demonstrate an appealing feature of uncertainty distillation; specifically, that for black-box models such as Gemini it is possible to achieve high performance better than semantic clustering with the efficiency of prompting, while most other accurate uncertainty quantification measures cannot be applied without open access to model weights.

## 8  CONCLUSION

**Findings**  We find that uncertainty distillation leads to improved estimates of uncertainty in comparison to many strong baselines, including baselines that require considerably more samples at inference-time. Additionally, we demonstrate that the representations of uncertainty learned during uncertainty distillation are applicable to unfamiliar test sets, showing that the model is learning to predict its own uncertainty independent of the subject of the dataset. Overall, we view our contribution as a significant step towards LLMs that can reliably reason about uncertainty, without requiring any auxiliary models or incurring additional inference-time compute.

**Future work**  While we focus on QA tasks, our method could be applied to tasks outside simple QA through the use of LLM verifiers to calculate binary correctness, as discussed in Appendix A. Future work may also investigate the robustness of the model's internal representation of uncertainty to even more dramatic domain shifts, such as different types of QA tasks or even tasks such as machine translation that bear no similarity to question answering. Looking beyond these immediate questions, LLMs that are able to verbalize meaningful confidences, for example thanks to our method, may be useful in a variety of applications requiring reasoning about uncertainty, such as medical diagnosis.

## LIMITATIONS

Our experiments focus on established QA tasks which admit straightforward ways to assess correctness. In principle, our approach generalizes to more complex tasks involving longer-form generations than the open-answer QA task described here; we leave it as future work to experiment in these settings with LLM verification. Separately, the proposed approach may be useful in cases where a single generation involves multiple distinct claims that each need to be associated with distinct confidences. Future work should identify appropriate datasets to evaluate multi-claim uncertainty estimation. We hope that our findings will encourage further study into uncertainty distillation in more general settings.

## REPRODUCIBILITY STATEMENT

We have endeavored to make reproducing our results straightforward. We describe our datasets, models, and metrics in detail in §4.1; we provide the prompts used in Appendix E; we provide the used hyperparameters in Appendix L and Appendix J; and we report the compute resources and dataset licensing in Appendix K. We plan to release our code upon publication.

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

## A  DISCUSSING SEMANTIC REPRESENTATIONS

In this paper, we generally focus on the relatively easy task of consolidating semantically similar answers for multiple-choice question answering datasets. In this case, semantic normalization is trivial, as it simply requires isolating the letter of the multiple-choice option, removing the reasoning and punctuation that affect lexical uncertainty quantification methods. However, for more complex tasks other approaches may be required (Huang et al., 2024). Previous research has established how normalization might be applied: for example, Kuhn et al. (2023) use natural language inference to cluster semantically equivalent answers and Tian et al. (2023) use an LLM as a judge of correctness. To demonstrate this variant, we set up an experiment to demonstrate how uncertainty distillation can be applied to an open-answer dataset.

### A.1  OPEN-ANSWER EXPERIMENTS

**Model and dataset**   We run these experiments with Llama-3B-Instruct. For the open dataset, we use GSM8K (Cobbe et al., 2021), an open math QA dataset consisting of grade-school level math problems. This dataset presents input variance that prevent exact match metrics from working effectively: even assuming the model correctly only encloses the final answer in the tags, an answer might be expressed as "10", "10 dollars", "$10", "10.00", and so on. All of these answers are semantically equivalent, but "10" would be the only accepted answer. We take the first 7000 examples of the training set as training data, the remaining 473 examples as validation data, and the existing test set as the unseen test set.

**Semantic normalization**   To make the clusters, we use code from Kuhn et al. (2023), specifically the EntailmentDeberta with minor changes to look for the absence of contradiction rather than entailment[19]. Once each sample has been generated, we compare answers pairwise, first to the correct answer (Formatted as "The correct answer is" followed by the simple numerical answer), and then to existing clusters. If none match, the answer is assigned to a new cluster. We choose a random answer to represent each cluster when constructing training data. The remainder of uncertainty distillation proceeds as normal.

---

[19]As Deberta (He et al., 2020) is trained for natural language inference, rather than comparing two numbers, absence of contradiction works better to cluster than entailment.

**Baselines and metrics** The baselines are described in §4. For P(IK), rather than using exact match to assign correctness labels to train the probe, we use EntailmentDeberta. At inference, to evaluate generated answers for all baselines, we query GPT-3.5-turbo as a judge.

**Results and analysis** We find that uncertainty distillation in this setting outperforms all baselines by a wide margin, achieving AUROC of 0.787. Both AUROC and high accuracy are significantly higher than the two baselines we compare to, and AUROC is similarly high to our multiple-choice question answering results, demonstrating that uncertainty distillation can be successfully applied to open-answer tasks by using semantic clustering to normalize answers at data generation.

| METHOD | AUROC | ACC | HIGH ACC | HIGH % |
|---|---|---|---|---|
| UD (OURS) | **0.787** | 0.752 | 0.935 | 58.0 |
| LEXICAL BASELINE | 0.542 | 0.829 | 0.832 | 98.2 |
| PROMPTING | 0.587 | 0.763 | 0.803 | 63.5 |

Table 4: AUROC and accuracy metrics for the open-answer experiments with Llama-3B-Instruct. We find that uncertainty distillation (UD) leads to increased AUROC and accuracy in high-confidence categories.

## A.2 GENERALIZATION TO LONG-FORM TASKS

For longer-form generation tasks, a single binary confidence judgment may be inadequate. The uncertainty distillation procedure is straightforward to extend to settings involving more than one prediction per generation, providing more granular feedback at the level of individual claims. As a concrete example, consider the task of extracting key facts from a news article. To apply our framework to this setting, we use a structured output consisting of a set of independent predictions, each of which associated with a *different* confidence. For a document from the CNN/DailyMail corpus[20], we show how `gemini-2.5-flash-lite` extracts key facts out-of-the-box using a suitable prompt and JSON output. However, as shown in §7, the calibration of the verbalized confidences is poor. By sampling multiple sets of claims for the same document, we can estimate the probability of each fact independently, and then apply our calibration and data augmentation procedure to fine-tune a model that outputs better calibrated confidences without incurring any additional runtime overhead. This requires using a semantic clustering approach which groups together equivalent factual statements expressed differently, which may be the same base model or a more efficient natural language inference model (Kuhn et al., 2023).

**Prompt**: Read the following document and summarize at most 5 key facts of the story as a list of claims (fewer is fine). Keep the claims short, simple, and in canonical format. Return your answer in JSON format with the field 'claims', which is a list of objects containing 'text' and 'confidence'.

**Document**: Solar flares can cause widespread disruption on Earth, but scientists have struggled to understand how they are formed. Now, a team from from the California Institute of Technology (Caltech) have taken a huge step forward by recreated the phenomenon that leads to solar flares in a lab. They hope their work could lead to improved space weather forecasts. The US team amanged to recreate plasma loops (above) in a lab for the first time [...]

```
{
  "claims": [
    {
      "text": "Scientists have recreated the phenomenon that leads
          to solar flares in a lab.",
      "confidence": "VERY HIGH"
    },
    {
      "text": "The laboratory recreation involved creating plasma
          loops, which are precursors to solar flares.",
      "confidence": "VERY HIGH"
```

---

[20]`abisee/cnn_dailymail`

```
10      },
11      {
12        "text": "The Caltech team hopes this research will lead to
             improved space weather forecasts and warnings.",
13        "confidence": "VERY HIGH"
14      },
15      {
16        "text": "Two magnetic forces were identified as controlling
             the behavior of the arching plasma loops.",
17        "confidence": "HIGH"
18      },
19      {
20        "text": "A novel technique using color-coded, inflowing
             plasma was employed to visualize the plasma loop
             behavior.",
21        "confidence": "HIGH"
22      }
23    ]
24  }
```

## B  TARGETED ANALYSIS

### B.1  EXPERIMENTAL SETTING

In §3, we assume that we have access to held-out calibration data. However, due to the unknown composition and size of pretraining datasets, it is increasingly challenging to guarantee that this assumption holds. We therefore test uncertainty distillation in a setting where we can know with certainty whether the calibration set is in a model's pretraining data. We examine this question and the impact of adding varying numbers of incorrect answers during uncertainty distillation in Appendix B.

**Dataset**   In this setting, we use the Super-NaturalInstructions dataset (SNI; Wang et al., 2022). We select 15 English Q&A tasks with short-form answers. We focus on Q&A tasks for which a single correct answer exists (e.g. multiple choice problems, short-form span extraction, math problems, etc.) and thus for which correctness of a model's prediction can reliably and efficiently be computed after normalizing lexical forms without resorting to methods such as LLM verification. We use 1,000 samples to obtain our Monte Carlo estimate of confidence (see Appendix D for details on how number of samples affects successful confidence estimation).

**Models**   We perform uncertainty distillation on FLAN-T5 (Chung et al., 2022), an instruction-tuned model trained on a dataset containing the SNI tasks. Importantly, we not only verify that Flan-T5 has been instruction-tuned on our tasks, but has seen samples from the *calibration set* of our test tasks. This allows us to investigate the effect of data contamination on calibration of verbalized confidences.

To construct a similar model which has *not* seen our calibration data, we instruction-tune a T5-Large model on a remaining subset of the English tasks in the SNI dataset, making sure to explicitly hold out the 15 tasks we use in our uncertainty distillation experiments. The result is an instruction-tuned model which we refer to as Instruct-T5, capable of performing our target Q&A tasks without having seen these tasks during training. In other words, the samples we obtain from this model do not require Instruct-T5 to be pre-trained on that specific task. See Appendix H for more details on our data selection and instruction-tuning. We train and evaluate uncertainty distillation on the combined dataset of these tasks and report the performance over the metrics described in §4.3.

**Baselines**   We report a comparison to the lexical baseline described above in order to provide context for the performance of the small models.

## B.2 RESULTS

**Assumption of calibration set**   We compare the performance of FLAN-T5, which has been instruction-tuned on the calibration set, with the performance of Instruct-T5, which has not, in Table 5. We find that while uncertainty distillation still produces meaningful confidence bins for FLAN-T5, it no longer outperforms lexical uncertainty. We conclude that uncertainty distillation works in the absence of held-out calibration data, but not as effectively as token-level probabilities, which are likely well-calibrated due to the model's previous training on these examples. We discuss results for these two models further in §B.2 and Appendix G, and find that the behavior of FLAN-T5 differs significantly from results on models where we have an unseen calibration set.

| MODEL | METHOD | AUROC | OVERALL ACCURACY | HIGH ACCURACY |
|-------|--------|-------|------------------|---------------|
| INSTRUCT-T5 | UNCERTAINTY DISTILLATION | **0.751** | **0.449** | **0.839** |
| | LEXICAL BASELINE | 0.667 | 0.387 | 0.754 |
| FLAN-T5 | UNCERTAINTY DISTILLATION | 0.873 | 0.614 | 0.875 |
| | LEXICAL BASELINE | **0.892** | **0.657** | **0.912** |

Table 5: AUROC and accuracy metrics when using FLAN-T5, which does not have an unseen calibration set. We find that while uncertainty distillation outperforms our lexical baseline with a model with an unseen calibration set, it does not outperform the baseline on FLAN-T5, which was instruction-tuned on the data previously.

**Adding incorrect examples**   While adding incorrect examples into the training data has the potential to provide more examples at different levels of confidences, it also is likely to increase the likelihood that a model generates an incorrect answer. To demonstrate this effect, in Table 6, we show the AUROC and accuracy for models trained with different amounts of incorrect samples. With Instruct-T5, we find that adding only two incorrect samples per correct sample dramatically increases AUROC while decreasing accuracy. While this would seem to indicate a fundamental tradeoff between accuracy and calibration, we find that the same is not as obviously true for FLAN-T5; while the accuracy may decrease and AUROC may increase, the effects are not as significant as they are for Instruct-T5. One possible interpretation of this is that its predictions are shaped by the fact that the data was included in its instruction-tuning corpus, leading to less dramatic shifts when trained.

While adding incorrect samples may improve AUROC, it increases the number of training examples by a factor of the number of incorrect examples added (e.g. a training set with 100 examples would train on 100 augmented answers with 0 incorrect examples added, 200 augmented answers with one incorrect example added, etc.) This leads to increased compute at training time. For this reason, in addition to the decreased accuracy, we recommend adding a low number of incorrect examples to the training dataset, and in our main experiments limit to at most one incorrect answer per question.

| | 0 | 1 | 2 | 3 |
|--------|-------|-------|-------|-------|
| INSTRUCT-T5 | | | | |
| AUROC | 0.723 | 0.737 | 0.751 | **0.757** |
| ACCURACY | **0.529** | 0.486 | 0.449 | 0.447 |
| FLAN-T5 | | | | |
| AUROC | 0.868 | 0.876 | 0.873 | **0.883** |
| ACCURACY | 0.609 | **0.620** | 0.614 | 0.611 |

Table 6: AUROC of models trained with varying numbers of incorrect examples allowed per question. There is a general trend towards increasing AUROC and decreasing accuracy when incorrect examples are included, although this is less pronounced for FLAN-T5.

## B.3 ANALYSIS

One high-level takeaway is that with small models there appears to be a tradeoff between an LLM's ability to predict its own confidence and overall model accuracy, but that this effect is less obvious with increasing model sizes. In our small-scale analysis, interventions that improve AUROC decrease accuracy and vice versa; however, with larger models we do not note as noticeable a decrease in accuracy compared to our baselines. Functionally, UD combines aspects of two tasks: the model's original question answering ability and uncertainty quantification. Large models are both less prone to catastrophic forgetting(Ramasesh et al., 2022)and more effective at multitask learning than smaller models(Chung et al., 2022). With this framing, the fact that larger models' accuracy is less impaired by the finetuning process of uncertainty distillation indicates that model scale plays a significant role in an accuracy/performance tradeoff, and increasing model scale or training in an explicitly multi-task setting may decrease the likelihood of drops in accuracy.

## C  UNCERTAINTY DISTILLATION ON SUPERVISED FINE-TUNED MODELS

We here examine uncertainty distillation's efficacy when performed on a small fine-tuned model, rather than large instruction-tuned models.

**Dataset**   We perform these experiments using the SQuAD benchmark (Rajpurkar, 2016). This is a machine-reading task where each question consists of a passage of text and one or more associated questions, each of which is answerable based on the text itself. As the test set has not been publicly released, we use the splits proposed by Du et al. (2017), which divides the publicly available available training and validation splits into train, test, and validation splits. We consider the first 60,000 examples in the training set to be training data, and the remainder to be our calibration set.

**Model**   We apply uncertainty distillation to T5-base (Raffel et al., 2020) finetuned on a portion of SQUAD. We use defaults for most hyperparameters, and report hyperparameters in Appendix L.

**Results**   Table 7 shows the results on the fine-tuned T5-base model. Uncertainty distillation achieves AUROC of 0.805 in the T5-base SQUAD experiment, slightly outperforming the lexical baseline's AUROC of 0.771.

| MODEL | METHOD | AUROC | OVERALL ACCURACY | HIGH ACCURACY |
|---|---|---|---|---|
| T5-BASE | UNCERTAINTY DISTILLATION | 0.805 | 0.711 | 0.852 |
|  | LEXICAL BASELINE | 0.771 | 0.811 | 0.865 |

Table 7: AUROC and accuracy metrics for T5-base, trained on SQUAD. We find that even in this setting, a model trained with uncertainty distillation outperforms lexical uncertainty in verbalizing confidences on SQUAD-T5

## D  NUMBER OF SAMPLES

Our Monte Carlo estimation of probability requires sampling repeatedly from a model before normalizing and calculating probability. In Figure 3, we show that the number of samples used to estimate the initial probabilities has a significant impact if chosen to be too low; however, there are diminishing returns as the number of samples increases. We therefore choose to use 1,000 samples in all of our experiments with FLAN-T5 and Instruct-T5, as more than that is unlikely to achieve anything but marginal improvement. For the larger models, we select 100 samples, as this appears to be the elbow of the curve in Figure 3, and as sampling 1000 samples from the large models would be computationally prohibitive. We note, however, that based on these results, this hyperparameter can be changed to improve efficiency or effectiveness of the method as is required by each task.

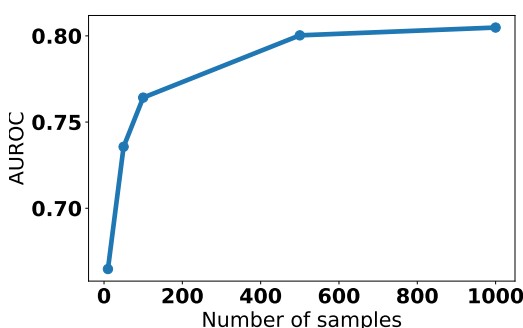

Figure 3: Curve showing the AUROC as a function of number of samples on the SQUAD dataset.

## E PROMPTS

### E.1 MISTRAL, LLAMA

**Prompt baselines, uncertainty distillation (MC)** Answer the following question and state confidence in the answer (very low, low, medium, high, very high). Enclose concise reasoning in <reasoning> </reasoning> tags, confidence in <confidence> </confidence> tags, and the letter of your FINAL answer in <answer> </answer> tags without any of your work, like this: "If each of Lisa's 7 chickens lays 6 eggs, how many eggs does Lisa have?
A) 24
B) 35
C) 42
D) 50
<reasoning> This can be solved with multiplication. The answer is 7*6, or 42.</reasoning> <answer> C) 42 </answer> <confidence>very high</confidence>." Your answer should not include words.

**Prompt baseline, uncertainty distillation (open)** "You are a helpful AI assistant. Answer the following math question as briefly as possible and accurately. Enclose confidence in the answer (very low, low, medium, high, very high) after the answer in <confidence> </confidence> tags, like so: <confidence> very high </confidence>."

**Sampling, lexical baseline** Answer the following question. Enclose concise reasoning in <reasoning> </reasoning> tags and the letter of your FINAL answer in <answer> </answer> tags without any of your work, like this: "If each of Lisa's 7 chickens lays 6 eggs, how many eggs does Lisa have?
A) 24
B) 35
C) 42
D) 50
<reasoning> This can be solved with multiplication. The answer is 7*6, or 42.</reasoning> <answer> C) 42 </answer>." Your answer should not include words.

**Sampling, lexical baseline (open)** "You are a helpful AI assistant. Answer the following math question as briefly as possible and accurately."

## E.2 INSTRUCT-T5, FLAN-T5

Each task in SNI has an associated instruction. For **sampling and the lexical baseline**, we simply use this instruction. For uncertainty distillation, we append ``Additionally state how confident you are in your answer'' to the instruction.

## E.3 LLM-AS-A-JUDGE

```
We are evaluating answers to the question {question}
Here are two possible answers:
Possible Answer 1: {text1}
Possible Answer 2: {text2}
Is Possible Answer 1 equivalent to Possible Answer 2, or do
the answers contradict?  Respond only with 'equivalent' or
'contradictory'.
```

## F BIN AND LABEL CHOICE

In the main experiments, we examine the effect of UD with five bins and a verbalized naming scheme. However, in Figure 4, we examine the effect of running UD on SocialIQA with Llama-3B while varying the number of bins (and thus necessarily changing the labeling scheme). Here, we find appropriate calibration regardless of number of bins. Notably, even changing the labeling scheme to numerical percentages does not result in a change in performance, suggesting that UD is robust to variance in labeling schemes. We use five bins as the default, as it offers enough bins to be challenging while avoiding the problems of sparsity (and thus noisiness) in bins that arise with larger bin sizes.

## G EFFECTS OF POST-HOC CALIBRATION

If the model's initial predictions are poorly calibrated, the post-hoc calibration step should help to better align probabilities in the training data with the true likelihood of success. In Table 8, we compare the miscalibration of the training data (measured through ECE with 30 bins) to the performance of models with and without post-hoc calibration during data generation. Unsurprisingly, we find that post-hoc calibration has positive effect corresponding to the initial miscalibration of the training data. For instance, Llama-3B on SocialIQA achieves 0.784 AUROC when trained on post-hoc calibrated data, and only 0.691 when identically trained on data without post-hoc calibration, with an ECE of 0.10. However, Ministral-8B on SocialIQA has a comparatively small ECE of 0.026, and the performance without post-hoc calibration is equivalent to the performance with post-hoc calibration. We conclude that the decision to include post-hoc calibration can be quickly and cheaply made by simply measuring the calibration of the annotated training data.

| DATASET | MODEL | WITH POST-HOC | NO POST-HOC | TRAINING DATA ECE |
|---------|-------|---------------|-------------|-------------------|
| MMLU | MINISTRAL-8B | **0.693** | 0.689 | 0.033 |
|      | LLAMA-3B | **0.743** | 0.714 | 0.039 |
| SIQA | MINISTRAL-8B | 0.671 | **0.673** | 0.026 |
|      | LLAMA-3B | **0.784** | 0.691 | 0.100 |

Table 8: AUROC of large models with and without post-hoc calibration at training time. We find that post-hoc calibration tends to improve performance, most dramatically with Llama-3B on SIQA.

We further analyze how post-hoc calibration impacts the model when small models are already well-calibrated on the specific task. Figure 5 shows the reliability diagrams for T5-base on SQUAD and Instruct-T5 on SNI. The models' predicted confidences align well with their actual accuracies; this allows us to investigate whether post-hoc calibration has a significant impact on AUROC for smaller models. Additionally, FLAN-T5 has been previously tuned on our calibration set; this gives us a setting to investigate the impact of post-hoc calibration when unseen calibration data is unavailable and the model is presumably correctly confident in its predictions.

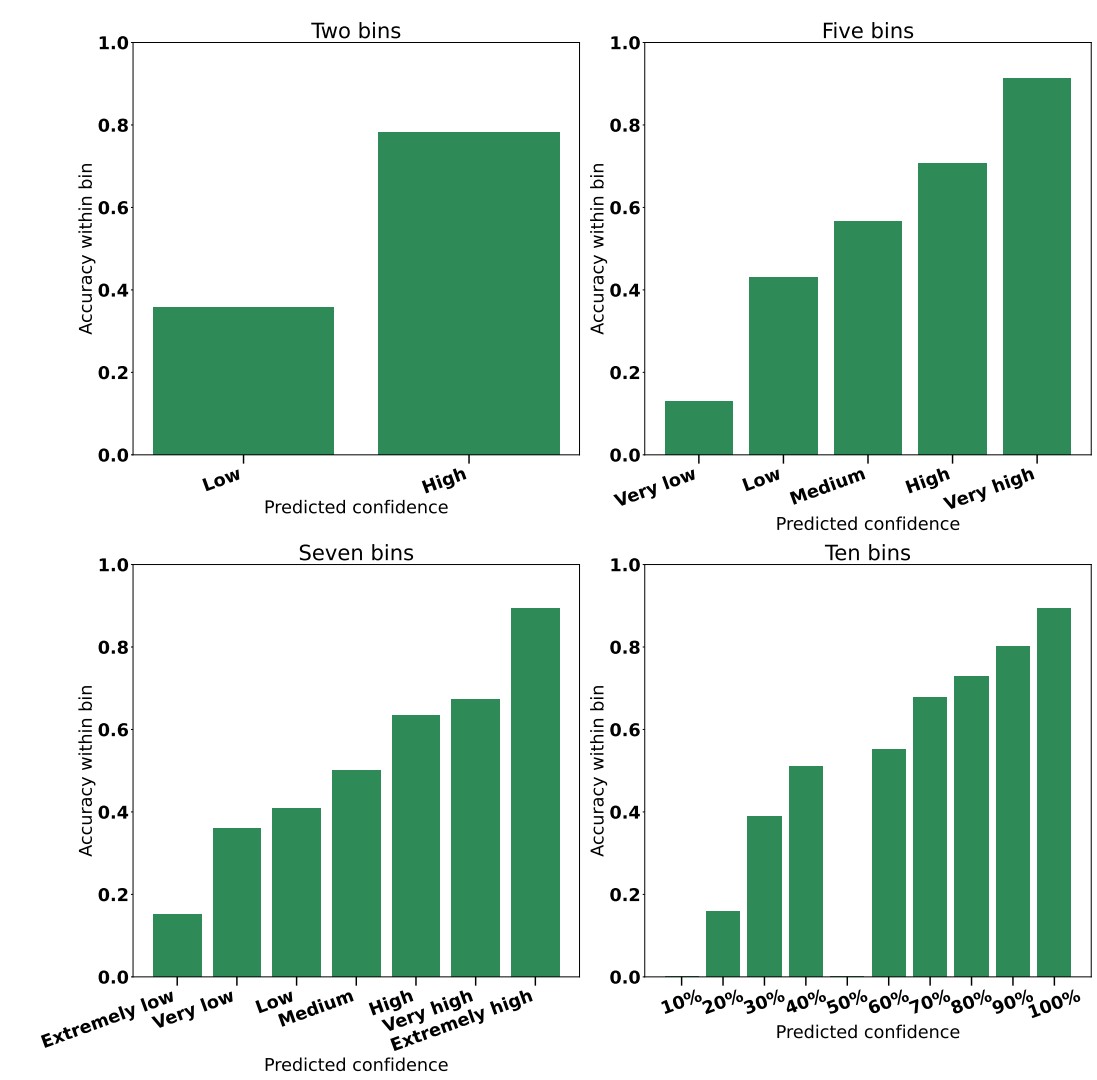

Figure 4: Running UD on SIQA with Llama-3B and changing the number of the bins, or the labeling scheme, has no noticeable effect on the efficacy of UD aside from increased sparsity in bins. As in other figures, bins with fewer than 10 samples are not plotted.

In Table 9, we show the results of the smaller models trained with and without this post-hoc calibration step. We find no apparent benefit of post-hoc calibration for Instruct-T5 or fine-tuned T5-base. These models are already well-calibrated on their domains; similarly to large models, a post-hoc calibrator does not significantly alter the output probabilities.

In the case of FLAN-T5, post-hoc calibration decreases AUROC. This suggests that in cases when unseen calibration data cannot be obtained for small models, uncertainty distillation may be more effective without the post-hoc calibration step.

## H SUPERNATURAL-INSTRUCTIONS TASKS

### H.1 TARGET CALIBRATION TASKS

As we describe in §3, in this work we rely on the assumption that our target-tasks have a correct answer, in the sense that it can be easily verified that an answer is right or wrong. Although this is not a strict necessity for calibration, it allows for us to define our buckets in terms of expected accuracy, rather than e.g. an expected score. We therefore focus on *short-form* Q&A tasks, question-

| DATASET | MODEL | WITH POST-HOC | NO POST-HOC |
|---------|-------|---------------|-------------|
| SQUAD | T5-BASE | 0.804 | 0.800 |
| SNI | INSTRUCT-T5 | 0.751 | 0.751 |
|  | FLAN-T5 | 0.873 | 0.883 |

Table 9: AUROC of well-calibrated models with and without post-hoc calibration at training time. We find that there is no notable performance increase with post-hoc calibration, and that there is a performance *decrease* when the model has previously been tuned on the calibration data.

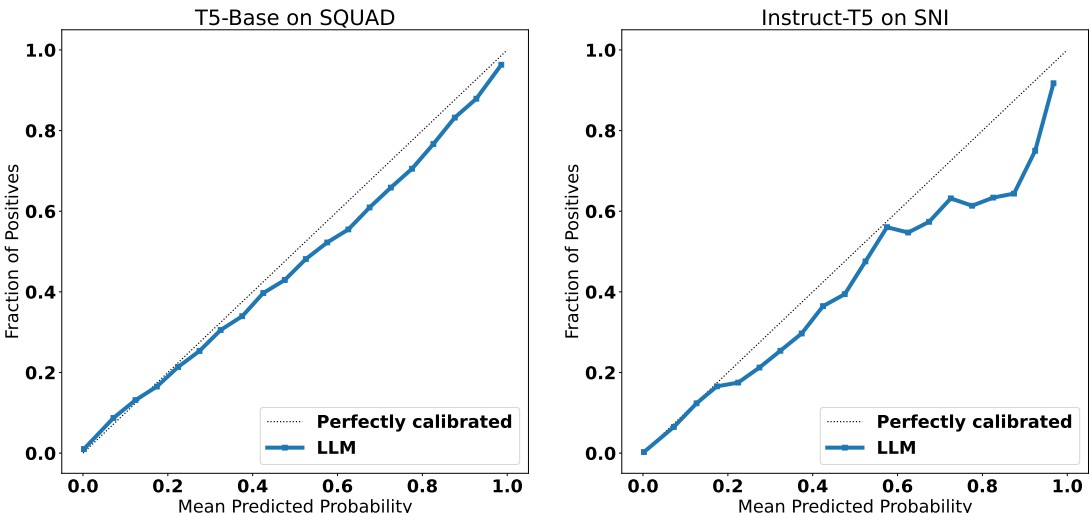

Figure 5: Initial calibration of our T5-base and Instruct-T5 model. Both models are well-calibrated in their respective domains, indicating that post-hoc calibration may not be necessary.

answer pairs whose answers consist of either selection from a fixed answer set (e.g. multiple choice or fixed choice) or single-word answers. We identify 15 tasks from the SuperNatural-Instructions dataset (Wang et al., 2022) that fit our criteria, and hold out these tasks as our uncertainty prediction tasks.

These tasks are split across 4 rough task types: **Multiple Choice** tasks involve selecting an answer from a set of choices, where the response is either a number or letter indicating the choice; **Fixed Choice** tasks involve selecting an answer from a pre-defined set of choices that are constant across the task (e.g. respond with either `True` or `False`); **Span Selection** tasks involve selecting the correct span of text from context and responding with that span as the answer; **Open Answer** involves generating the answer to the question in an open-ended way, i.e. the answer is not provided in the context.

For all tasks, we ensure that the answers are no more than 2 words long, making it easy to perform normalization and verify accuracy for each question. The tasks are shown in Table 10; for each task, we use 10% of the samples as a validation set, 10% of the samples as a held-out test set, use the remaining 80% of the data to form our calibration set.

## H.2 INSTRUCTION-TUNING TASKS

Because most modern instruction-tuned models are trained on all of Super-NaturalInstructions, they have seen the our calibration target tasks during instruction-tuning. Therefore, we instruction-tune our own T5 model to test the effectiveness of our method on unseen tasks. Our model is trained on a subset of the SuperNatural-Instructions dataset (Wang et al., 2022). Specifically, we instruction-tune on the English split used in the original paper but we take out our target calibration tasks identified in §H.1. This gives us a training dataset of 879 instruction-tuning tasks, with a total of roughly 1.2M training samples total.

| Task Type | Task Name |
|---|---|
| Multiple Choice | `task580-socialiqa-answer-generation`
`task309-race-answer-generation`
`task1297-qasc-question-answering`
`task1420-mathqa-general`
`task228-arc-answer-generation-easy`
`task1286-openbookqa-question-answering`
`task1431-head-qa-answer-generation`
`task1731-quartz-question-answering`
`task750-aqua-multiple-choice-answering` |
| Fixed Choice | `task380-boolq-yes-no-question`
`task1661-super-glue-classification` |
| Span Selection | `task002-quoref-answer-generation`
`task041-qasc-answer-generation` |
| Open Answer | `task591-sciq-answer-generation`
`task898-freebase-qa-answer-generation` |

Table 10: The tasks and task types that we select from the SuperNatural-Instructions dataset for validating and testing our calibration method.

To validate our models instruction-following capabilities, we use the in-context learning test set from SuperNatural-Instructions, which contains 95 additional held out tasks from task categories that are not seen in the training dataset.

# I    MINISTRAL PLOTS

In Figure 6 we display the plots with Ministral-8B. As reflected in the AUROC score in Table 1, calibration is slightly worse; however, compared to baselines, it still does a more accurate job of forecasting accuracy.

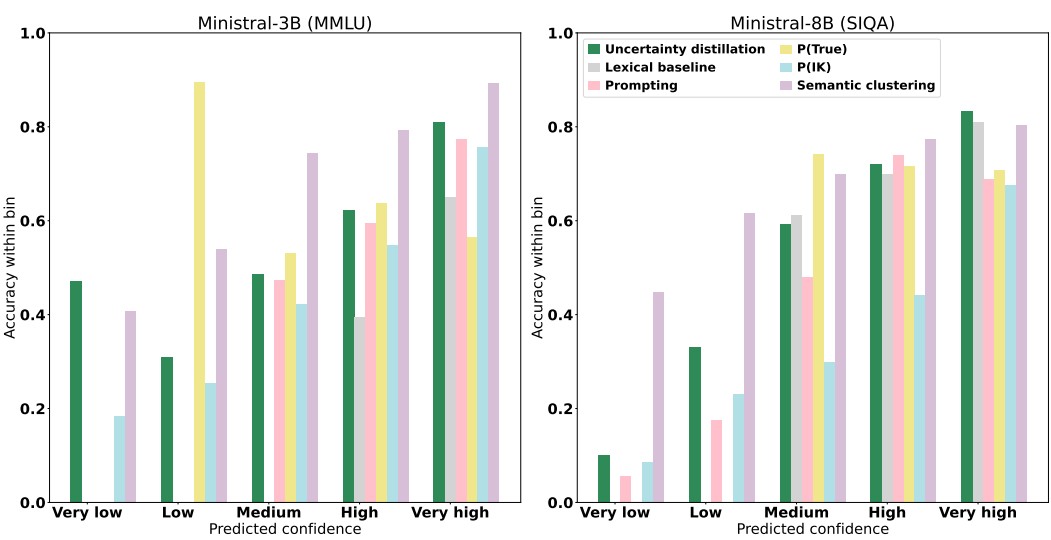

Figure 6: Average accuracy within each confidence bin for our main experiments. We do not plot bins with fewer than 10 samples.

## J INSTRUCTION-TUNING T5

We follow a standard recipe for instruction-tuning T5-Large, established in Wang et al. (2022). Specifically, we tune the model for 3 epochs with a batch size of 16 and a learning rate of $5 \times 10^{-3}$. We use the AdamW optimizer, and a constant learning rate schedule after a warmup period of 500 steps. During instruction-tuning, we train the model with the semantic definition of each task prepended to the task input, and we similarly prompt the model when performing our target Q&A tasks.

## K RESOURCE REPORTING

### K.1 COMPUTE RESOURCES

Here we report the compute resources used in this work. Instruction-tuning T5 took a total of 200 GPU hours across 4 NVIDIA-V100s. Running uncertainty distillation on Instruct-T5 and FLAN-T5 took 16 hours per model on a single NVIDIA-H100. Finetuning T5-base on SQUAD for our initial model took 3 hours on a single NVIDIA RTX 2080, and training using uncertainty distillation took 8 hours on a single NVIDIA-V100. Finetuning Ministral-8B (LoRA) and finetuning Llama-3B each took took three hours on two NVIDIA-A100s. Our lexical baseline for SQUAD took one hour on one NVIDIA RTX 2080; for SNI took three hours on one NVIDIA RTX 2080; for MMLU took three hours on one NVIDIA-A100; for SocialIQA took three hours on one NVIDIA-A100; for GSM8K took four hours on one NVIDIA-A100. Prompting for MMLU and prompting for SocialIQA took 1 hour on one NVIDIA-A100. Sampling for SQUAD took a total of 60 GPU hours on NVIDIA-V100s; for SNI took 45 GPU hours on NVIDIA-A100s; for SocialIQA took 350 hours on NVIDIA-A100s; for MMLU took 350 hours on NVIDIA-A100s; for GSM8k took 80 hours on NVIDIA-A100s.

### K.2 RESOURCE INTENDED USE

Super-NaturalInstructions (SNI) is an open-source instruction tuning dataset, released under the Apache License.[21] The intended use of SNI is to instruction-tune language models to learn to follow instructions, and to evaluate a model's ability to follow instructions on unseen tasks. While we use the SNI dataset for precisely this purpose during instruction-tuning, we also use 15 held-out tasks to serve as uncertainty quantification tasks. This does not necessarily fall under the intended use of instruction-tuning; however, the authors of SNI also mention that the dataset may serve as a large, multi-task natural language resource (Wang et al., 2022), and our usage of the target calibration tasks does fall under this use case.

The Stanford Question Answering Dataset (SQUAD) (Rajpurkar, 2016) is distributed under the Creative Commons Attribution-Sharealike 4.0 license, which permits use of the dataset as long as it is properly attributed and as long as the results are distributed under the same license. As we cite the paper and plan to publically release our code and models after acceptance, our use of this dataset is permitted under this license.

SocialIQA (Sap et al., 2019) is not explicitly licensed, but they state that they " establish Social IQa as a resource" for future models.

MMLU (Hendrycks et al., 2020a), GSM8K(Cobbe et al., 2021), and MMLU-pro(Wang et al., 2024b) are published under the MIT license, which allows users to freely copy, use, and change the licensed material.

## L UNCERTAINTY DISTILLATION HYPERPARAMETERS

In Table 11 and Table 12, we show the training hyperparameters for uncertainty distillation training. All experiments in Table 11 added two incorrect answers per question, and in Table 12 added one incorrect answer per question.

---

[21]Available here: `https://github.com/allenai/natural-instructions`

**Hyperparameters for fine-tuning via API**  For the experiments reported in §7, we fine-tune the `gemini-2.5-flash-lite` model using LoRA with rank 4 for 10 epochs and defaults for other hyperparameters. On validation data, we compared performance for different numbers of incorrect examples (§3.3), finding that augmenting the tuning set with a single incorrect prediction had marginal impact on accuracy while significantly improving calibration. We also compared both temperature scaling and isotonic regression, finding that isotonic scaling produced better calibration, while temperature scaling produced higher accuracy. To fit the calibration map, we held out 10% of the training data.

| Model | Epochs | Learning rate | Batch size | Grad accum steps |
|---|---|---|---|---|
| T5-base (initial) | 1 | 3e-5 | 12 | 1 |
| T5-base (Uncertainty distillation) | 3 | 3e-5 | 12 | 1 |
| Instruct-T5 (Uncertainty distillation) | 3 | 3e-5 | 1 | 32 |
| FLAN-T5 (Uncertainty distillation) | 3 | 3e-5 | 1 | 32 |

Table 11: Hyperparameters for training all T5 models but Instruct-T5 (see Appendix J for details). All models are trained with the AdamW optimizer.

| Model | Epochs | Learning rate | Batch size | LoRA rank | LoRA alpha |
|---|---|---|---|---|---|
| Llama-3B/MMLU | 3 | 4e-5 | 4 | - | - |
| Llama-3B/SocialIQA | 1 | 3e-5 | 4 | - | - |
| Ministral-8B/MMLU | 3 | 5e-5 | 4 | 16 | 32 |
| Ministral-8B/SocialIQA | 1 | 3e-5 | 4 | 8 | 16 |
| Llama-3B/GSM8K | 1 | 8e-6 | 4 | - | - |

Table 12: Hyperparameters for training all Llama and Ministral models. Gradient accumulation steps is 1 for each model. All models are trained with the AdamW optimizer.

# M  ALGORITHM

**Algorithm 1** Uncertainty distillation

---

**Require:** Language model $f_\theta$ with params $\theta_0$
**Require:** Calibration set $S^{cal} = \{X^{cal}, Y^{cal}\}$
  $S^{scored} \leftarrow \emptyset$
  **for** $(x, y) \in S^{cal}$ **do**
    $D \leftarrow \{\hat{y}_i\}_{i=1}^N \sim f_\theta(x)$
    Normalize $D$ by semantics, and count
    **for** $\hat{y} \in D$ with count $n$ **do**
      $f \leftarrow \frac{n}{N}$
      $S^{scored} \leftarrow S^{scored} \cup \{(x, \hat{y}, y, f)\}$
    **end for**
  **end for**
  $c() \leftarrow \texttt{isotonic\_regression}(S^{scored})$
  $S^{vc} = \emptyset$
  **for** $(x, \hat{y}, y, f) \in S^{scored}$ **do**
    **if** $\texttt{filter}(\hat{y}, y)$ **then**
      $\texttt{continue}$
    **end if**
    $p \leftarrow c(f)$
    $b \leftarrow \texttt{bin}(p)$
    $z \leftarrow \texttt{verbalize\_confidence\_map}(\hat{y}, b)$
    $S^{vc} \leftarrow S^{vc} \cup \{(x, z)\}$
  **end for**
  $\mathcal{L}(\theta) \leftarrow \mathbb{E}_{(x,z) \in S^{vc}}[NLL(f_\theta(x), z)]$
  $\theta_{cal} \leftarrow \texttt{train}(\theta_0, \mathcal{L})$
  Return $\theta_{cal}$

---

