# OpenReview forum: "Uncertainty Distillation: Teaching Language Models to Express Semantic Confidence"
_ICLR.cc/2026/Conference — Submitted to ICLR 2026_

### Official Review · Reviewer_vCfX · 2025-10-15

**Soundness:** 3
**Presentation:** 3
**Contribution:** 2
**Rating:** 4
**Confidence:** 3

**Summary:**

This paper proposes an uncertainty distillation method, which teaches the model to express its uncertainty to the answer through sft using a dataset with confidence labels collected through Monte Carlo sampling. Compared to sampling-based semantic UQ methods the proposed method has better inference-time efficiency. Experiments show that uncertainty distillation generally achieves higher AUROC scores and works well on OOD scenarios.

**Strengths:**

- Paper is clear and easy to follow

- The proposed method makes sense in theory and shows good performance under the predefined setup.

- The experiment is comprehensive an convincing.

**Weaknesses:**

- Practical Issue: The proposed method requires a significant amount of Monte Carlo samples of the training data responses, while the performance under more significant distribution drift from training to test is not studied. This may limit the practicability of the proposed method due to the significant computation overhead. The post-hoc calibration in Section 3.2 post an additional constraint to the answer format, i.e., it is only applicable to closed-form solutions such multi-choice QA, which further reduce the use cases of this method.

- Baselines: I'm not very familiar with verbalized uncertainty but it looks like there are some more similar baselines such as [1]. Some more efficient methods mentioned in the "Lexical uncertainty quantification" paragraph in Related Works can also be used as baselines, since the authors are testing on open source models.

- Minor Correctness Issue: The authors' claim in footnote 11 about not adopting ECE is, IMO, not precise. First of all, ECE does not require continuous probability. In addition, it has only one hyperparameter---the number of bins. But the authors present calibration plot in Figure 2 anyways so I do not find this issue significant.

- Reproducibility: no code or data available.


[1] Liu, Shudong, et al. "Can llms learn uncertainty on their own? expressing uncertainty effectively in a self-training manner." Proceedings of the 2024 Conference on Empirical Methods in Natural Language Processing. 2024.

**Questions:**

See above

---

> ### Author Response · Authors · 2025-11-27
> **Response to Reviewer vCfx**
>
> Thank you for your close reading of our paper, and for the actionable feedback that you gave us, which we believe has improved the quality of our paper. To address the main concerns you articulated:
>
> >” The proposed method requires a significant amount of Monte Carlo samples of the training data responses…”
>
> Using a relatively expensive offline process for the purpose of training a fast online model is not uncommon in deep learning: take, for instance, model distillation. Concretely, for Gemini 2.5 (Flash), the total cost of our post-training procedure is around $20. For this cost, we obtain a model that achieves higher AUROC than semantic clustering (for 1/20th the inference time cost, for the recommended 20 sample version of semantic clustering). For production settings where a large number of customers need to be served, the relatively modest post-training cost of our method would massively reduce ongoing scaling costs.
>
> >” the performance under more significant distribution drift from training to test is not studied…”
>
> We believe that the distribution drift from training a model on social commonsense questions and asking about (for instance) science, math, and law, is already fairly significant and demonstrates that UD is robust to reasonable shifts. Additionally, we further validate our approach   in section 7 on MMLU-Pro, which covers a wide range of topics and covers both reasoning and knowledge questions. Our success tuning Gemini 2.5 (Flash) for this task, using only a few hundred examples from each area, suggests that a single model fine-tuned using our approach can provide consistently good uncertainty estimates across many tasks.
>
> >”...it is only applicable to closed-form solutions such multi-choice QA…”
>
> We previously noted in Appendix A and limitations that methods such as semantic clustering (see Kuhn et al., 2023), which have been shown empirically to work well to normalize answers, could straightforwardly be applied to open-ended tasks. To demonstrate this point, we have now added an experiment on open-ended question answering in Appendix A.1. We find that using semantic clustering to normalize the training data does not demonstrate a significant degradation in performance, and believe this provides empirical evidence that there is a concrete solution for open-ended tasks.
>
> >”...ECE does not require continuous probability. In addition, it has only one hyperparameter…”
>
> We appreciate this correction and have rephrased to elaborate. While ECE may not require continuous probability, it requires a numerical probability, which we do not possess for our discrete confidence bins. Assigning a probability has a high impact on calibration error: for instance, assigning 0% accuracy vs 20% accuracy would be equally valid for “very low”, as this is the range that we convert to “very low”, but would result in very different calibration error depending on the choice. Rather than deciding arbitrarily, we present the calibration plots to visually show calibration. The hyperparameters we are referencing come from Measuring Calibration in Deep Learning by Nixon et al. This paper describes other hyperparameters that may be used for calibration error, such as binning strategy or type of regularization.
>
> >”Reproducibility: no code or data available.”
>
> We intend to publish our code publicly and have edited the paper to note as much in the text. We have also described our datasets in detail, although we plan to also publish code to replicate our splits. In the interim, we make a comprehensive and documented implementation of our procedure for fine-tuning via API available. As noted in the updated main text, our results on MMLU-Pro can be reproduced using the provided repository (page 2) for ~$20 in a few hours without specialized hardware.
> We believe that this response should address the major concerns you identified in this paper, as well as demonstrating that UD can be applied to more diverse models and tasks than previously established. Please let us know if there are additional concerns or revisions you would like us to address.

---

### Official Review · Reviewer_svHf · 2025-10-28

**Soundness:** 2
**Presentation:** 3
**Contribution:** 2
**Rating:** 4
**Confidence:** 3

**Summary:**

This paper proposes uncertainty distillation, a fine-tuning approach to teach LLMs to verbalize calibrated semantic confidence in their answers. The method uses Monte Carlo sampling with semantic normalization to estimate answer probabilities, applies post-hoc calibration via isotonic regression, and fine-tunes models to output verbalized confidence levels alongside predictions. Evaluated on MMLU and SocialIQA with in-domain and out-of-domain transfer scenarios, the approach achieves competitive or better AUROC compared to stronger baselines, particularly semantic clustering methods, with negligible inference-time overhead.

**Strengths:**

The paper addresses a problem: models are often confidently incorrect, and enabling them to express calibrated uncertainty has significant practical value. The three-step approach is straightforward and doesn't require architectural changes or external models at inference time, making it immediately applicable to existing models.

A major practical advantage is efficiency. Unlike semantic clustering methods requiring 20 inference passes, this approach generates a single response with verbalized confidence, maintaining minimal inference-time overhead. The method is rigorous in its experimental design with proper in-domain and out-of-domain evaluation, comprehensive ablations on including incorrect examples, and careful dataset splits preventing data contamination. Results are evaluated on models ranging from small (FLAN-T5, T5-base) to medium (Llama-3B, Ministral-8B).

The paper reveals interesting findings: there's a calibration-accuracy trade-off where smaller models suffer more when trained on incorrect examples, and the approach shows better robustness to domain shift than supervised baselines. The comparison between Instruct-T5 (unseen during pre-training) versus FLAN-T5 (potentially contaminated) effectively illuminates which performance gains are genuine. The presentation is clear with effective figures showing calibration quality across confidence bins.

**Weaknesses:**

The evaluation is restricted to multiple-choice QA with only two datasets (MMLU and SocialIQA). There's no evaluation on generation tasks like summarization, machine translation, or code generation, nor on open-ended tasks where semantic normalization becomes challenging. Model evaluation is limited to parameters ≤8B, leaving unclear how the approach scales to larger models.

Semantic normalization is a fundamental limitation. The current approach (isolating answer letters, removing punctuation) works for multiple-choice QA but is inherently task-specific. The appendix acknowledges this is "trivial" for short-form QA but "challenging" for complex generation tasks, yet provides no concrete solution. This severely restricts generalizability beyond standardized QA formats.

A critical practical issue is data contamination. FLAN-T5 results show substantial performance degradation when the calibration set appears in pre-training data, raising questions about real-world applicability. The paper doesn't address how to identify unseen calibration data for closed-source models, making deployment in production environments uncertain.

The theoretical understanding is limited. Why does this particular combination of techniques improve calibration? The connection between semantic probability estimates and actual accuracy remains unclear. Design choices like the five-bin verbalization scheme and specific confidence labels lack principled justification, and the paper doesn't discuss when the method might fail or produce misleading outputs.

Several methodological concerns undermine confidence in the results. The choice of 100 Monte Carlo samples appears arbitrary with no principled justification.Additionally, the offline requirement for collecting and annotating calibration data is a practical burden. There's a potential overfitting issue: post-hoc calibration is applied on the calibration set and evaluated on the same set. Table 1 shows accuracy drops in some cases (Llama-3B MMLU AUROC improves but accuracy decreases), but when this trade-off is acceptable isn't clearly discussed. High-confidence predictions are often sparse (34.6% for Ministral on MMLU), limiting practical utility.

**Questions:**

1. Beyond QA, how would you apply semantic normalization for summarization or translation tasks?

2. For FLAN-T5, why does post-hoc calibration actually hurt performance (Table 6)?

3. What's the optimal number of Monte Carlo samples as a function of model size?

4. Can you provide deeper analysis of the accuracy-calibration trade-off?

5. How does the method perform with more fine-grained confidence levels (7-10 categories vs. 5)?

---

> ### Author Response · Authors · 2025-11-27
> **Response to Reviewer svHf**
>
> Thank you for your detailed critique, which suggested multiple new analyses that we could run to strengthen our paper. We address your concerns below.
>
> >”Semantic normalization is a fundamental limitation...yet provides no concrete solution…”
>
> We have noted that Kuhn et al., 2023 has established methods of semantic normalization, which have been shown empirically to work well to normalize answers and can straightforwardly be applied to open-ended tasks. However, we had previously left this to future work, and acknowledge the importance of demonstrating that this can be effectively combined with uncertainty distillation. To demonstrate this point, we have now added an experiment on open-ended question answering in Appendix A, and have clarified how this could be extended to long-form tasks such as summarization on a toy example with Gemini-2.5. Empirically, we find that using semantic clustering to normalize the training data does not demonstrate a significant degradation in performance, and believe this provides empirical evidence our approach provides a concrete solution for open-ended tasks.
>
> >”...unclear how the approach scales to larger models…”
>
> We note that we have demonstrated the efficacy of uncertainty distillation at a variety of model sizes, with performance remaining effective regardless of size. However, to further demonstrate that UD can be used for large models, we have added experiments using Google’s Gemini 2.5 model in section 7 , which vastly outscales any of the open models we have run with. While the architecture of Gemini 2.5 has not been publicly disclosed, some sources estimate it to be over 100 billion parameters. We find that UD outperforms baselines even for Gemini, which we believe should offset the concern that the approach would not scale.
>
> >”substantial performance degradation when the calibration set appears in pre-training data….how to identify unseen calibration data for closed-source models”
>
> We acknowledge that there was some performance degradation of FLAN-T5 when post-hoc calibration was used; however, we note that a decrease in AUROC from 0.883 to 0.873 does not represent “substantial” performance degradation. However, to address this concern, we have conducted further analysis in Appendix G, comparing our results with Llama and Ministral to models trained without post-hoc calibration on our two datasets. These results, combined with the calibration error of the training data, show that it is possible to determine how much of an effect post-hoc calibration will have by simply measuring the calibration of the training data. If the training data is already well-calibrated (which may occur because the training data appears in pre-training data), it is simple to leave out the post-hoc calibration step. We thank the reviewer for prompting this analysis, which we believe addresses the concern of determining whether to use post-hoc calibration when automatically creating data to deploy UD in production environments.
>
> >”...the five-bin verbalization scheme and specific confidence labels lack principled justification…”
>
> One benefit of the fine-tuning step is that the number of bins and method of verbalizing confidence are within the control of the user; In Appendix F we have added an analysis where we vary the number of bins and labels for Llama-3B on SocialIQA. This analysis suggests changing the number of bins or the way confidence is verbalized does not significantly affect the success of uncertainty distillation beyond increasing sparsity in bins. We chose to use five bins as it demonstrated the ability to differentiate on a relatively fine-grained scale (i.e. beyond “likely correct” or “likely incorrect”) while assigning a small enough number of labels to decrease the chance that the probability in any particular bin is influenced by sparsity.
>
> >”the choice of 100 Monte Carlo samples appears arbitrary with no principled justification”
>
> We apologize for not elaborating on our justification, and have modified the paper to more explicitly describe the logic. We referenced experiments with smaller models to decide on the number of samples, as described in Appendix D. To balance the need for performance with the computational expense of sampling large models, we picked 100 samples as the evident elbow of the curve. However, we also note that should more efficiency be required, a user could take only 50 samples, or could increase to 500 samples should precision be important. As Figure 3 demonstrates, performance varies by amount of samples, but the method remains somewhat effective with as few as 10 samples.

---

> ### Author Response · Authors · 2025-11-27
> **Response to Reviewer svHf (cont)**
>
> >”the offline requirement for collecting and annotating calibration data is a practical burden”
>
> We acknowledge that automatically constructing our training data is computationally expensive. However, using an expensive offline process for the purpose of training a fast online model is not uncommon in deep learning: take, for instance, model distillation. As you point out, our model maintains minimal inference-time overhead, compared to methods such as semantic clustering which require the same cost at inference time. For Gemini 2.5 (flash) the total cost of our post-training procedure is around $20.  For this cost, we obtain a model that achieves higher AUROC than semantic clustering (for 1/20th the inference time cost, for the recommended 20 sample version of semantic clustering). For production settings where a large number of customers need to be served, the relatively modest post-training cost of our method would massively reduce ongoing scaling costs.
>
> >”Beyond QA, how would you apply semantic normalization for summarization or translation tasks?”
>
> In Appendix A.2, we provide a demonstration using Gemini-2.5 of how semantic clustering as normalization could be applied to tasks requiring longer-form generation, such as summarization or long-form MT. Concretely, the semantic normalization from Appendix A.1 can be applied to subclaims in the generation; extracted facts, in the case of summarization. We provide an example of this for summarization below:
>
> Document: “Solar flares can cause widespread disruption on Earth, but scientists have struggled to understand how they are formed. Now, a team from from the California Institute of Technology (Caltech) have taken a huge step forward by recreated the phenomenon that leads to solar flares in a lab. They hope their work could lead to improved space weather forecasts. The US team amanged to recreate plasma loops (above) in a lab for the first time [...]”
> ```{
>   "claims": [
>     {
>       "text": "Scientists have recreated the phenomenon that leads to solar flares in a lab.",
>       "confidence": "VERY HIGH"
>     },
>     {
>       "text": "The laboratory recreation involved creating plasma loops, which are precursors to solar flares.",
>       "confidence": "VERY HIGH"
>     },
>     {
>       "text": "The Caltech team hopes this research will lead to improved space weather forecasts and warnings.",
>       "confidence": "VERY HIGH"
>     },
>     {
>       "text": "Two magnetic forces were identified as controlling the behavior of the arching plasma loops.",
>       "confidence": "HIGH"
>     },
>     {
>       "text": "A novel technique using color-coded, inflowing plasma was employed to visualize the plasma loop behavior.",
>       "confidence": "HIGH"
>     }
>   ]
> }
> ```
>
>  >” post-hoc calibration is applied on the calibration set and evaluated on the same set…”
>
> The post-hoc calibrated data that is used to train our model is entirely separate from the test sets we use to evaluate AUROC, so there should not be overfitting. We have clarified this in the paper more explicitly in section 3.3.
>
> >”when this trade-off is acceptable isn't clearly discussed”
>
> We generally find that for large models including Llama, Mistral, and Gemini-2.5, UD results in accuracies that are increasingly close to the range of accuracies achieved by our baselines. For instance, while Llama-3B MMLU’s accuracy is reduced compared to prompting, it is still higher than the lexical baseline. However, we do note accuracy tradeoffs for the smaller models, and therefore agree that this trade-off should be explicitly discussed, which we do in Appendix B.3. We believe that a key takeaway is that accuracy is notably more affected for small models than larger models (notably, accuracy seems unaffected for Gemini, as we discuss in section 7), and that accuracy drops appear to become less prevalent as model size increases; this may indicate that improved multitask performance explains the difference in accuracy performance, as UD is functionally training two tasks at once.

---

> > ### Author Response · Authors · 2025-11-27
> > **Response to Reviewer svHf (cont)**
> >
> > >”High-confidence predictions are often sparse (34.6% for Ministral on MMLU)...”
> >
> > We apologize for a poorly-structured sentence which we believe caused this misunderstanding. The sentence in question referenced the two cases where uncertainty distillation did not have the highest “high accuracy”, reading “In both these cases, the high accuracy improvement comes at the cost of a notably smaller percentage of samples in high-confidence bins, with only 34.6% of predictions being high-confidence in the first case and only 26.4% of predictions being high-confidence in the second, compared to 49.7% and 55.1% respectively for uncertainty distillation.” We have rephrased this sentence to better clarify that 34.6% and 26.4% refer to the baselines, rather than our method. However, we will also note that the number of high-confidence predictions necessarily will and should vary depending on the model’s knowledge of the task. In high-uncertainty situations, having more than 50% of samples be classed as high confidence may be misleading. We report these numbers not as a marker of success, but to demonstrate that the high accuracy metrics are not “gaming the system” by selecting a trivial number of examples for UD.
> >
> > We believe that our updated experiments have addressed most of the concerns articulated in your review, and thank you once again, as we believe these analyses have significantly contributed to the strength of our method. If you have further comments, we are happy to continue to revise.

---

### Official Review · Reviewer_m3xZ · 2025-11-03

**Soundness:** 3
**Presentation:** 3
**Contribution:** 2
**Rating:** 4
**Confidence:** 4

**Summary:**

SFT is used to make models quantify their uncertainty directly using generated tokens after giving an answer to factual questions. Using a Monte Carlo sampling approach, semantically equivalent answers all get corresponding values. There is much less compute overhead during inference than you would get for a fully blackbox method, at the cost of being able to retrain the model.

**Strengths:**

Regarding the semantic/lexical/verbalized uncertainty quantification, the definitions make sense and are a good summary of existing black-box approaches. The test for domain shift is an important question for such trained methods, and strengthens the empiric section of this work.

**Weaknesses:**

More than strongly beating all baselines, this paper seems to propose a tradeoff. Better semantic confidence, in some cases, which comes at the cost of some model performance. Extra tokens (very few) have to be generated, but not as many as in sampling based methods. Fine tuning on the other hand requires some sampling, but not access to activation at inference time (I would here note that unlike what is stated line 092, fine tuning does access model weights - even if pragmatics of that paragraph imply authors mean direct access during inference, I would advise rephrasing). Overall the tradeoff in compute in terms of memory, and time could be more systematic.

Regarding the semantic/lexical/verbalized uncertainty quantification, I do not see metrics of performance on input variation. Is this something you expect the monte Carlo sampling to take care of? I do not see an empirical verification of it's success. Perhaps something along the lines of the Kuhn et al 2023, or a model generated approach Mahaut et al 2024 could be relevant to make explicit the success of your method, beyond performance. This seems to be a major advantage of the method.

Typo and minor:
There seems to be a missing space in the footnote line 214
I find the use of the word DISTILLATION confusing. Distillation is already applied in specific setting of model compression and teacher/student model - applying it here leads to extra confusion in my opinion.

**Questions:**

See Weaknesses. Additionally, more of a personal curiosity question : why the ISOTONIC regression specifically here?

---

> ### Author Response · Authors · 2025-11-27
> **Response to Reviewer m3xZ**
>
> Thank you for the excellent feedback on our paper. To discuss the main concerns:
>
> >”...which comes at the cost of some model performance…”
>
> While we note an accuracy tradeoff in our controlled studies with T5, we also note that this effect lessens when model size increases. In the case of Llama and Ministral, there is only one case where our model achieves the lowest accuracy of all uncertainty quantification methods, and in the case of our new Gemini experiments there is no notable decrease in accuracy. Overall, with current generation models, our approach maintains accuracy with equivalent baselines, while significantly improving calibration.
>
> >”...unlike what is stated line 092, fine tuning does access model weights…”
>
> We appreciate this distinction and have rephrased this sentence to “white-box” access to model weights, as fine-tuning through APIs does not reveal model weights. Our intent was to convey that our method can be applied to models such as OpenAI’s GPT family or Google’s Gemini family of models, which will not output token probabilities (thus making token-level uncertainty quantification impossible) or model activations (making probing methods such as P(IK) impossible). We demonstrate that this approach is effective in new experiments in Section 7, and compare it to prompting and semantic clustering, the only two baselines that could similarly be applied to Gemini. We find that our method vastly outperforms prompting, and achieves improved AUROC over semantic clustering (even when giving semantic clustering *32 times* the number of samples) with a single sample. We believe this demonstrates how UD has broader use cases than many of our baselines.
>
> >”...the tradeoff in compute in terms of memory…”
>
> Could you clarify where the tradeoff in memory exists? To our knowledge, there is no noticeable change in memory.
>
> >”... I do not see metrics of performance on input variation…something along the lines of the Kuhn et al 2023…”
>
> In the case of multiple choice sampling, we used a simple standardization. However, the methods you describe would work to standardize input variation, as we previously discussed in limitations and appendix A. We have run additional experiments with Llama-3B on GSM8k to show empirically that using techniques from Kuhn et al 2023 works in Appendix A.
>
> >”... I find the use of the word DISTILLATION confusing…”
>
> We chose to use the phrase “uncertainty distillation” to suggest a connection with model distillation; in both cases, data is generated through a computationally expensive process and a more efficient model learns to replicate the high-quality predictions. For example, in Hinton’s paper, an expensive ensemble is distilled into a single model by generating samples from the ensemble. While our procedure is different, the notion of distillation is the same. We have edited the paper to include a footnote explaining this connection.
>
> >”...why the ISOTONIC regression specifically here?”
>
> We chose to use isotonic regression as it is a commonly used calibration method; however, the method would work with other calibration methods, or even (less effectively) without a post-hoc calibration method. We have added a note to explain this. In Appendix L, we also note the effect of temperature scaling as a post-hoc calibration method for Gemini, which achieves marginally lower AUROC with marginally higher accuracy.
>
> Once again, we thank you sincerely for the time taken to read and understand the paper. We are happy to discuss any remaining concerns further.

---

### Author Response · Authors · 2025-11-27
**Overall Response**

We thank the reviewers for their careful reading of the paper. We appreciate the general consensus that our paper presents a straightforward approach with demonstrable empirical success even under noticeable domain shifts. We respond to individual reviewer concerns in specific responses. An updated version of the paper has been uploaded addressing reviewer feedback; changes to the paper have been written in blue text for easy comparison, and we present here a summary of the most major changes to the paper:

1. **Applying UD to black-box models**: in section 7 we demonstrate that uncertainty distillation can be effectively applied to Gemini, a black-box model that allows API fine-tuning. This highlights that uncertainty distillation works with very large models and demonstrates a distinguishing feature of our approach compared to prior work. We make a comprehensive and documented implementation of our procedure for fine-tuning via API available in an anonymous repository (see Page 2). As noted in the updated main text, our results can be reproduced using this repository for ~$20 in a few hours without specialized hardware.
2. **Applying UD to open answer tasks**: in Appendix A we demonstrate our previously-suggested extension of semantic normalization at data generation to show how uncertainty distillation can be applied to tasks that do not use string normalization to normalize answers. We find that UD can be effectively used for our open-answer dataset, and that it is straightforward to apply semantic normalization as described by Kuhn et al. (2023). We additionally describe a straightforward generalization of our framework to longer-form tasks such as summarization, by producing multiple claims each associated with separate confidence statements.
3. **In-depth analysis of post-hoc calibration**:  In Appendix G we show that performance improvements caused by post-hoc calibration at training time correlate with miscalibration of training data. This experiment clarifies why data contamination may have previously correlated with degradation in efficacy of post-hoc calibration, and shows a practical and efficient method for determining whether or not to use post-hoc calibration when automatically creating training data.
4. **Experimenting with number of bins and confidence labels**: In Appendix F we compare Llama-3B’s performance with differing numbers of bins and different confidence labels, finding that uncertainty distillation is effective regardless of number of bins or confidence labels.
5. **General writing changes**: We have edited the writing at the suggestion of reviewers for improved clarity.
6. **Minor bug fix**: We found that an incorrect prompt was used when training Ministral-8B/SocialIQA. We have updated the tables with the corrected numbers, which show a minor improvement in AUROC for our method.

---

### Meta-Review · Area_Chair_4B6C · 2026-01-06

**Summary:**

This paper proposes uncertainty distillation, a fine-tuning approach that trains language models to output calibrated, verbalized confidence alongside their answers.
Reviewers agreed that the problem is important and that the approach is simple, well-motivated, and efficient at inference compared to sampling-based semantic uncertainty methods. The main concerns raised were the practical usage. These included the cost and complexity of the offline data construction, limited evaluation beyond multiple-choice question answering, questions about scalability to larger or black-box models, and the observed trade-off between calibration quality and task accuracy. Reviewers also raised questions about design choices such as the number of Monte Carlo samples, confidence bins, and post-hoc calibration, as well as reproducibility.
The rebuttal and revised paper addressed many of these concerns with additional experiments and clarifications.

**Reviewer Concerns:**

### Concerns addressed by the rebuttal:

- The authors added experiments on open-ended question answering and clarified how existing semantic normalization methods can be applied beyond multiple-choice settings.

- New results on Gemini 2.5 demonstrated that the method can scale to very large, API-only models.

- The offline cost of Monte Carlo sampling was clarified and quantified, and framed as a one-time cost that trades off against much lower inference-time overhead.

- The role of post-hoc calibration was clarified, including analysis showing when it is likely to help or hurt performance.


### Concerns that remain outstanding:

- The requirement for offline sampling and calibration data remains a practical limitation, particularly for frequent retraining or rapidly changing domains.

- The calibration–accuracy trade-off persists for smaller models, and its acceptability depends on the application.

- Some ablations are evaluated only on limited examples.

**Reviewer Scores:**

Reviewer m3xZ: Initially4. Given the added experiments and clearer discussion of trade-offs, this reviewer may increase the score.

Reviewer svHf: Initially 4. Most concerns were addressed, especially the misunderstanding; this reviewer may increse the score.

Reviewer vCfX: Initially 4. Some of the questions are not fully addressed, such as the selection of baselines, and the review may keep the original rating.

---

### Decision · Program_Chairs · 2026-01-26

Reject